# Bias assessment of a test-negative design study of COVID-19 vaccine effectiveness used in national policymaking

Sophie Graham [1,2,3] ✉, Elise Tessier[2], Julia Stowe[2], Jamie Lopez Bernal[2], Edward P. K. Parker [1], Dorothea Nitsch [1,4,5], Elizabeth Miller[1,3], Nick Andrews[2,3], Jemma L. Walker[1,2,3,6] & Helen I. McDonald[1,3,6]

National test-negative-case-control (TNCC) studies are used to monitor COVID-19 vaccine effectiveness in the UK. A questionnaire was sent to participants from the first published TNCC COVID-19 vaccine effectiveness study conducted by the UK Health Security Agency, to assess for potential biases and changes in behaviour related to vaccination. The original study included symptomatic adults aged ≥70 years testing for COVID-19 between 08/12/2020 and 21/02/2021. A questionnaire was sent to cases and controls tested from 1–21 February 2021. In this study, 8648 individuals responded to the questionnaire (36.5% response). Using information from the questionnaire to produce a combined estimate that accounted for all potential biases decreased the original vaccine effectiveness estimate after two doses of BNT162b2 from 88% (95% CI: 79–94%) to 85% (95% CI: 68–94%). Self-reported behaviour demonstrated minimal evidence of riskier behaviour after vaccination. These findings offer reassurance to policy makers and clinicians making decisions based on COVID-19 vaccine effectiveness TNCC studies.

Test-negative-case-control (TNCC) observational studies are an important tool in the COVID-19 pandemic to monitor the continued real-world vaccine effectiveness of COVID-19 vaccinations against new variants and to assess the duration of protection[1–6]. In this design symptomatic individuals who present for testing for COVID-19 are included, categorised as cases if testing positive for COVID-19 and controls if testing negative. The design controls for confounding from health-seeking behaviour and healthcare access to some extent since both cases and controls are required to have accessed healthcare for COVID-19-like symptoms[7]. The design also controls for exposure because cases and controls have reported respiratory symptoms.

The UK Health Security Agency (UKHSA) has conducted regular COVID-19 vaccine effectiveness analyses in England using the TNCC design since vaccines were introduced in the UK in December 2020.

The first published study included individuals in England aged ≥70 years who had a COVID-19 test in the community with self-reported symptoms and a symptom onset date between 8th December 2020 and 21st February 2021[4]. Patients were excluded if they had a history of a previous positive COVID-19 test from 26th October 2020 until 7th December 2020 to ensure vaccine effectiveness was assessed in those more likely to be susceptible. The study found that from 14 days from a second dose of BNT162b2 and from 14–20 days after a first dose of ChAdOx1 (i.e., the available COVID-19 vaccinations at the time), vaccine effectiveness reached 89% (95% confidence interval [CI]: 85–93%) and 60% (95% CI 41–73%), respectively, which was in line with vaccine efficacy estimates from clinical trials[8,9]. The work from this study informed governmental policy at the time[4,10] and the subsequent analyses have been used to provide regular updated estimates for national policy-makers[6].

[1]London School of Hygiene and Tropical Medicine, London, UK. [2]UK Health Security Agency, London, UK. [3]National Institute for Health and Care Research (NIHR) Health Protection Research Unit in Vaccines and Immunisation, London, UK. [4]UK Renal Registry, Bristol, UK. [5]Renal Unit, Royal Free London NHS Foundation Trust, Hertfordshire, UK. [6]These authors contributed equally: Jemma L. Walker, Helen I. McDonald. ✉e-mail: sophie.graham@lshtm.ac.uk

Although the TNCC design aims to control for confounding from the opportunity to be exposed, health-seeking behaviour and healthcare access, it does not implicitly control for other confounders of vaccine effectiveness and these need to be accounted for in the analysis. In the aforementioned UKHSA COVID-19 vaccine effectiveness study, it was only possible to adjust for potential confounders that were available in the national vaccination (National Immunisation Management Service (NIMS)) and COVID-19 testing (Second Generation Surveillance System (SGSS)) datasets that were utilised. Although this dataset includes some sociodemographic information such as age, gender, geographical region, index of multiple deprivation (IMD) and care home status, other key potential confounders such as detailed information on comorbidities[11] and household type[12] could not be identified in this dataset at the time (Fig. S1).

Riskier behaviour during or after vaccination may also result in real-world vaccine effectiveness estimates that are lower than the efficacy observed in randomised placebo-controlled clinical trials[13]. For example, during the national lockdowns, individuals who knew they were vaccinated may have assumed they were protected and might have therefore mixed more with individuals outside their household which would have increased their likelihood of exposure to SARS-CoV-2. They also might have had an additional risk of exposure compared to non-vaccinated individuals whilst travelling to or from, or even at, vaccination centres (Fig. S1).

The current study used a questionnaire that was sent out to a subsample of the original UKHSA TNCC COVID-19 vaccine effectiveness study in individuals aged 70 years and over. The aim of the current study was to use the questionnaire data to attempt to quantify the size and direction of potential biases that may have impacted estimates from one of the first UK COVID-19 vaccine effectiveness studies.

## Results
### Population description and selection bias
Among the 23713 individuals that made up the questionnaire sample, 8648 (36.5%) responded to the questionnaire ("respondents") and 15065 (63.5%) did not respond ("non-respondents"; Fig. 1). Among respondents, self-reported history of COVID-19 vaccination (one or two doses) at the time of questionnaire completion was high (Table S1).

Amongst the 8648 respondents, there were 6741 vaccinated (BNT162b2 = 3531 and ChAdOx1-S = 3210) and 1907 non-vaccinated at symptom onset date (based on SGSS onset date). Amongst the 8648 respondents there were 6541 negative controls and 2107 cases. When comparing respondents with non-respondents of the questionnaire there did appear to be some demographic and clinical differences, with respondents being younger, more likely to be of White ethnicity,

less likely to live in a deprived area, and more likely to be a case when compared with non-respondents (based on a percentage absolute difference of +/−5% and p values of <0.05; Table S2). However, this selection bias did not appear to alter vaccine effectiveness estimates as after 2 doses of BNT162b2 vaccine, vaccine effectiveness in respondents (88% [95% CI: 79–94%]) was similar to non-respondents (87% [95% CI: 79–93%]) and the overall questionnaire sample (86% [95% CI: 79–91%]) (Fig. 2). There was insufficient follow-up to assess the effectiveness of two doses of ChAdOx1 vaccination, however, respondents had similar vaccine effectiveness from the first dose (14 days postvaccination) than non-respondents (Table S3).

Respondents that were vaccinated were more likely to be a negative case, were older and were more likely to have later testing and symptoms compared with respondents that were non-vaccinated. Respondents that were cases were more likely to be non-vaccinated, more likely to be from the Northeast and Yorkshire and less likely to be from the Southwest of England, more likely to be deprived, and were more likely to have earlier testing compared with respondents that were negative controls (based on +/−5% percentage absolute difference and p values of <0.05; Table 1).

The results for the potential biases and alternative causal pathways in the original TNCC are detailed below and summarised in Fig. 3 and Table S4.

### Potential biases in original TNCC study
When asking individuals in the questionnaire to report their vaccination date and comparing it to NIMS for the assessment of exposure misclassification, 9.5% (499/5276) of individuals reported a first dose vaccination date that was later in the questionnaire, whereas 7.3% (386/5276) reported a date that was earlier in the questionnaire (Fig. 3A). The remaining 83.2% (4391/5276) individuals reported the same date in NIMS and the questionnaire. The same pattern was seen for second-dose vaccinations. 89.8% of first doses self-reported in the questionnaire were within 3 days +/- of NIMS date (inclusive), whereas 3.6% were more than 3 days earlier and 6.6% were more than 3 days later in the questionnaire. For the second dose, 93.3% of self-reported vaccination dates were within 3 days +/− NIMS date, whereas 1.0% were more than 3 days earlier and 5.8% were more than 3 days later (Fig. S2A, B).

When updating vaccination dates to those self-reported in the questionnaire (or if missing using vaccination dates from NIMS), the percentage of individuals identified as non-vaccinated at symptom onset date (using SGSS) was very similar to when using NIMS (NIMS vaccination date: 22.1%; self-reported vaccination date: 23.5%). Vaccine effectiveness after two doses of BNT162b2 decreased from 88% (95% CI: 79–94%) to 84% (95% CI: 74–92%; Fig. 2). When exploring key

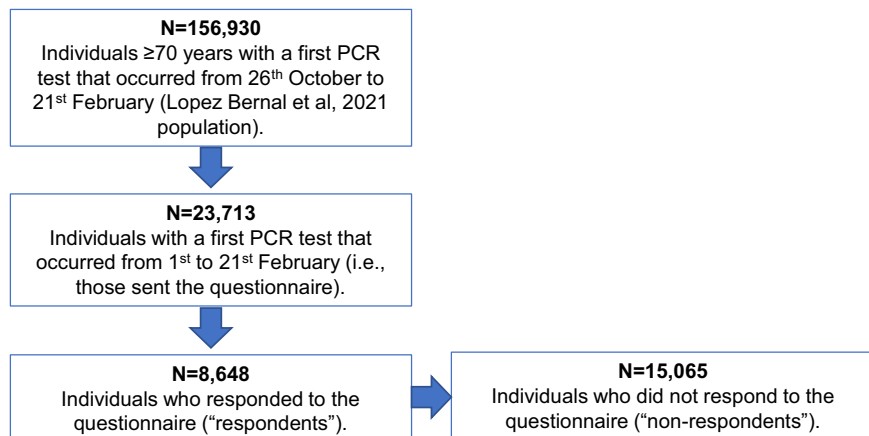

**Fig. 1 | Cohort Selection.** Abbreviations: PCR: polymerase chain reaction.

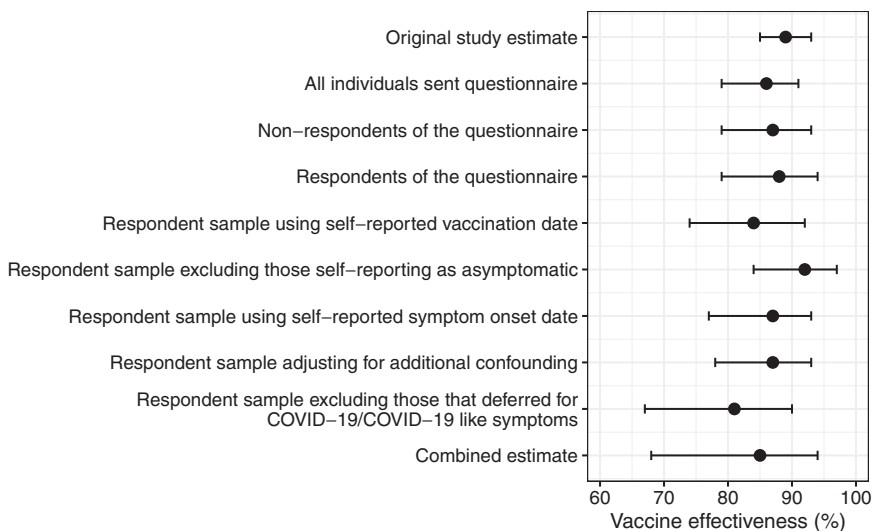

**Fig. 2 | Forest Plot: Vaccine effectiveness estimates after two doses of BNT162b2.** Vaccine effectiveness estimates after 2 doses of BNT162b2 in the following populations: the original TNCC study sample; all individuals who were sent the questionnaire; non-respondents of the questionnaire; respondents of the questionnaire; respondents using self-reported vaccination status; respondents using self-reported onset dates; respondents excluding those self-reporting as asymptomatic; respondents adjusting for additional confounding (for CEV, household size, and household type); respondents excluding those that delayed their vaccination because they had COVID-19/COVID-19 like symptoms; a combined estimate that accounted for all the above potential biases. All estimates adjusted for confounders that were adjusted for in the original TNCC study (age, gender, ethnicity, geography, index of multiple deprivations, care home status, and week of onset). Points represent odds ratio with corresponding 95% confidence intervals.

confounders amongst those with different self-reported vaccination status at symptom onset (from SGSS) versus unchanged status, we found that increased self-reported dose counts were associated with aged 75–79 years, most deprived IMD quintile, and COVID-19 symptom onset date in February week 1 but less associated with age 70–74 years (based on +/−5% percentage absolute difference and p values of <0.05; Table S5). On the other hand, decreased self-reported dose counts were associated with male gender, COVID-19 symptoms testing in February week 2 and were less associated with age 70–74 years, the 4th quintile of deprivation (with 5th being the lowest), and COVID-19 symptoms in February week 3 (based on +/−5% percentage absolute difference and p-values of <0.05; Table S5).

When asking individuals in the questionnaire to report their symptomatic status and comparing to SGSS for the assessment of outcome misclassification through symptomatic status, 65.5% (5539/8459) of the total population responding to this question reported that they were symptomatic despite all reporting they were symptomatic at the time of requesting their PCR test. Self-reported symptoms were 64.7% (4375/6741) and 67.4% (1285/1907) in vaccinated and non-vaccinated individuals, and 59.7% (3905/6541) and 83.5% (1759/2107) in negative controls and cases (Fig. 3B). When restricting to individuals who reported they had symptoms in the questionnaire vaccine effectiveness vaccine effectiveness for two doses of BNT162b2 increased from 88% (95% CI: 79–94%) in respondents of the questionnaire to 92% (95% CI: 84–97%; Fig. 2). Self-reported asymptomatic status was associated with older age and male sex, but no other key confounders (based on +/−5% percentage absolute difference and p-values of <0.05; Table S6).

When asking individuals in the questionnaire to report their symptom onset date and comparing to SGSS for the assessment of outcome misclassification through symptom onset date, 5.9% (514/8645) of individuals reported an earlier date, whereas 2.2% (194/8645) of individuals reported a later date in the questionnaire (Fig. 3C). 95.8% of these were within 3 days +/- of SGSS date (inclusive), whereas 3.0% were more than 3 days earlier and 1.2% were more than three days later in the questionnaire (Fig. S3).

When updating vaccination dates using self-reported onset dates, the percentage of non-vaccinated was very similar to when using SGSS (SGSS onset: 22.1%; self-reported onset: 22.6%). Vaccine effectiveness after two doses of BNT162b2 decreased marginally from 88% (95% CI: 79–94%) to 87% (95% CI: 77–93%; Fig. 2). The prevalence of confounders did not differ among individuals with differing versus unchanged self-reported symptom onset date (based on +/−5% percentage absolute difference and p-values of <0.05; Table S7).

For the assessment of confounding, when adjusting for COVID-19 risk factors self-reported in the questionnaire (household size, household type and CEV) in addition to the variables in the original study (age, gender, ethnicity, geography, index of multiple deprivations, care home status and week of onset), vaccine effectiveness estimates after two doses of BNT162b2 decreased marginally from 88% (95% CI: 79–94%) to 87% (95% CI: 78–93%; Fig. 2). Due to the later approval of ChAdOx1-S, there were insufficient individuals with two doses at symptom onset date (N = 5) for the same assessment to be made.

Individuals with COVID-19-like symptoms, recent exposure to COVID-19 or a positive SARS-CoV-2 test just before their vaccination date were recommended to defer their vaccination by 28 days according to government guidelines[14]. This deferral has the potential to increase vaccine effectiveness estimates as individuals that defer their vaccination, for this reason, might go on to test positive for SARS-CoV-2 (inflating cases among non-vaccinated individuals). In the current study, among all individuals who reported in the questionnaire that they had been vaccinated at the time of survey (8518/8613 98.9%), 9.6% (794/8251) delayed their vaccination ≥4 weeks from the invitation. Among these individuals, 25.3% (201/794) reported they delayed vaccination because of COVID-19/COVID-19 symptoms. Of all individuals who reported in the questionnaire that they had not been vaccinated (95/8613 1.2%), over a quarter (26/95 27.4%) of individuals reported that this was because they had been unwell or because they had COVID-19 (Fig. 3D and Table S2). Thus, some individuals appeared to be deferring their vaccinations because they were unwell or because they had COVID-19. When assessing for the potential impact of deferral bias, amongst those who didn't delay vaccination because of COVID-19/COVID-19 like symptoms (N = 8396), vaccine effectiveness after two doses of BNT162b2 decreased from 88% (95% CI: 79–94%) to 81% (95%

**Table 1 | Baseline characteristics of respondents, by vaccination and case status, using variables from the original study data (NIMS and SGSS)**

| Characteristics according to NIMS and SGSS | Respondents not vaccinated at symptom onset, N = 1907 | Respondents vaccinated at symptom onset, N = 6741 | Difference in absolute percentage (vaccinated – non-vaccinated) | p-value | Respondents who were negative controls, N = 6541 | Respondents who were cases, N = 2107 | Difference in absolute percentage (cases – negative controls) | p-value |
|---|---|---|---|---|---|---|---|---|
| **Vaccine status at symptom onset, n (%)** | | | | | | | | <0.001 |
| Not vaccinated | | | | | 1351 (20.7%) | 556 (26.4%) | 5.70% | |
| Vaccinated | | | | | 5190 (79.3%) | 1551 (73.6%) | −5.70% | |
| **Test result** | | | | <0.001 | | | | |
| Negative | 1351 (70.8%) | 5190 (77.0%) | 6.2% | | | | | |
| Positive | 556 (29.2%) | 1551 (23.0%) | −6.2% | | | | | |
| **Age group in years, n (%)** | | | | <0.001 | | | | 0.626 |
| 70–74 | 1492 (78.2%) | 2931 (43.5%) | −34.7% | | 3348 (51.2%) | 1075 (51.0%) | −0.20% | |
| 75–79 | 279 (14.6%) | 2056 (30.5%) | 15.9% | | 1778 (27.2%) | 557 (26.4%) | −0.80% | |
| 80–84 | 57 (3.0%) | 1031 (15.3%) | 12.3% | | 826 (12.6%) | 262 (12.4%) | −0.20% | |
| 85–89 | 43 (2.3%) | 473 (7.0%) | 4.7% | | 381 (5.8%) | 135 (6.4%) | 0.60% | |
| =>90 | 36 (1.9%) | 250 (3.7%) | 1.8% | | 208 (3.2%) | 78 (3.7%) | 0.50% | |
| **Gender, n (%)** | | | | 0.421 | | | | 0.115 |
| Female | 1081 (56.7%) | 3749 (55.6%) | −1.1% | | 3685 (56.3%) | 1145 (54.3%) | −2.00% | |
| Male | 826 (43.3%) | 2992 (44.4%) | 1.1% | | 2856 (43.7%) | 962 (45.7%) | 2.00% | |
| **Ethnicity, n (%)** | | | | 0.075 | | | | <0.001 |
| White | 1756 (92.1%) | 6266 (93.0%) | 0.9% | | 6114 (93.5%) | 1908 (90.6%) | −2.90% | |
| Non-white | 84 (4.4%) | 224 (3.3%) | −1.1% | | 185 (2.8%) | 123 (5.8%) | 3.00% | |
| Prefer not to say | 67 (3.5%) | 251 (3.7%) | 0.2% | | 242 (3.7%) | 76 (3.6%) | −0.10% | |
| **Geographical region, n (%)** | | | | <0.001 | | | | <0.001 |
| East of England | 246 (12.9%) | 814 (12.1%) | −0.8% | | 834 (12.8%) | 226 (10.7%) | −2.10% | |
| London | 104 (5.5%) | 614 (9.1%) | 3.6% | | 530 (8.1%) | 188 (8.9%) | 0.80% | |
| Midlands | 360 (18.9%) | 1415 (21.0%) | 2.1% | | 1265 (19.3%) | 510 (24.2%) | 4.90% | |
| Northeast and Yorkshire | 317 (16.6%) | 1043 (15.5%) | −1.1% | | 948 (14.5%) | 412 (19.6%) | 5.10% | |
| Northwest | 232 (12.2%) | 994 (14.7%) | 2.5% | | 922 (14.1%) | 304 (14.4%) | 0.30% | |
| Southeast | 394 (20.7%) | 1116 (16.6%) | −4.1% | | 1202 (18.4%) | 308 (14.6%) | −3.80% | |
| Southwest | 254 (13.3%) | 745 (11.1%) | −2.2% | | 840 (12.8%) | 159 (7.5%) | −5.30% | |
| **IMD quintile, n (%)** | | | | 0.085 | | | | <0.001 |
| 1 (most deprived) | 238 (12.5%) | 800 / 6736 (11.9%) | −0.6% | | 683 / 6536 (10.4%) | 355 (16.8%) | 6.40% | |
| 2 | 319 (16.7%) | 1018 / 6736 (15.1%) | −1.6% | | 963 / 6536 (14.7%) | 374 (17.8%) | 3.10% | |
| 3 | 399 (20.9%) | 1425 / 6736 (21.2%) | 0.3% | | 1379 / 6536 (21.1%) | 445 (21.1%) | 0.00% | |
| 4 | 477 (25.0%) | 1622 / 6736 (24.1%) | −0.9% | | 1645 / 6536 (25.2%) | 454 (21.5%) | −3.70% | |
| 5 (least deprived) | 474 (24.9%) | 1871 / 6736 (27.8%) | 2.9% | | 1866 / 6536 (28.5%) | 479 (22.7%) | −5.80% | |
| Missing | 0 | 5 | | | 5 | 0 | | |
| **Week of symptom onset, n (%)** | | | | <0.001 | | | | <0.001 |
| January week 1 | 10 (0.5%) | <5 | | | 10 (0.2%) | <5 | | |
| January week 2 | 32 (1.7%) | <5 | | | 35 (0.5%) | <5 | | |
| January week 3 | 86 (4.5%) | 61 (0.9%) | −3.6% | | 110 (1.7%) | 37 (1.8%) | 0.10% | |
| January week 4 | 737 (38.6%) | 987 (14.6%) | −24.0% | | 1238 (18.9%) | 486 (23.1%) | 4.20% | |
| February week 1 | 802 (42.1%) | 2202 (32.73%) | −9.4% | | 2203 (33.7%) | 801 (38.0%) | 4.30% | |
| February week 2 | 178 (9.3%) | 2202 (32.7%) | 23.4% | | 1839 (28.1%) | 541 (25.7%) | −2.40% | |
| February week 3 | 62 (3.3%) | 1280 (19.0%) | 15.7% | | 1106 (16.9%) | 236 (11.2%) | −5.70% | |
| **Week of COVID-19 test, n (%)** | | | | <0.001 | | | | <0.001 |
| February week 1 | 1385 (72.6%) | 2166 (32.1%) | −40.5% | | 2572 (39.3%) | 979 (46.5%) | 7.20% | |
| February week 2 | 362 (19.0%) | 2038 (30.2%) | 11.2% | | 1772 (27.1%) | 628 (29.8%) | 2.70% | |

**Table 1 (continued) | Baseline characteristics of respondents, by vaccination and case status, using variables from the original study data (NIMS and SGSS)**

| Characteristics according to NIMS and SGSS | Respondents not vaccinated at symptom onset, N = 1907 | Respondents vaccinated at symptom onset, N = 6741 | Difference in absolute percentage (vaccinated – non-vaccinated) | p-value | Respondents who were negative controls, N = 6541 | Respondents who were cases, N = 2107 | Difference in absolute percentage (cases – negative controls) | p-value |
|---|---|---|---|---|---|---|---|---|
| February week 3 | 160 (8.4%) | 2537 (37.6%) | 29.2% | | 2197 (33.6%) | 500 (23.7%) | −9.90% | |
| **Care home status, n (%)** | | | | 0.059 | | | | <0.001 |
| Not care home | 1901 (99.7%) | 6691 (99.3%) | −0.4% | | 6514 (99.6%) | 2078 (98.6%) | −1.00% | |
| Care home[a] | 6 (0.3%) | 50 (0.7%) | 0.4% | | 27 (0.4%) | 29 (1.4%) | 1.00% | |
| **CEV, n (%)** | | | | <0.001 | | | | 0.006 |
| Not CEV | 1709 (89.6%) | 5746 (85.2%) | −4.4% | | 5601 (85.6%) | 1854 (88.0%) | 2.40% | |
| CEV | 198 (10.4%) | 995 (14.8%) | 4.4% | | 940 (14.4%) | 253 (12.0%) | −2.40% | |

Abbreviations: *CEV* clinically extremely vulnerable, *IQR* interquartile range, *IMD* index of multiple deprivations, *n* numerator *N* denominator, *NIMS* National Immunisation Management System, *SGSS* Second Generation Surveillance System.

[a]Care home status is likely low in the current study because the study only included those tested in the community (pillar 2), individuals tested in care homes or in hospital are usually tested under pillar 1.

Note: all tests were conducted using two-sided Chi squared test.

Note: cells <5 have been suppressed and secondary suppression has also been conducted in order to protect patient privacy.

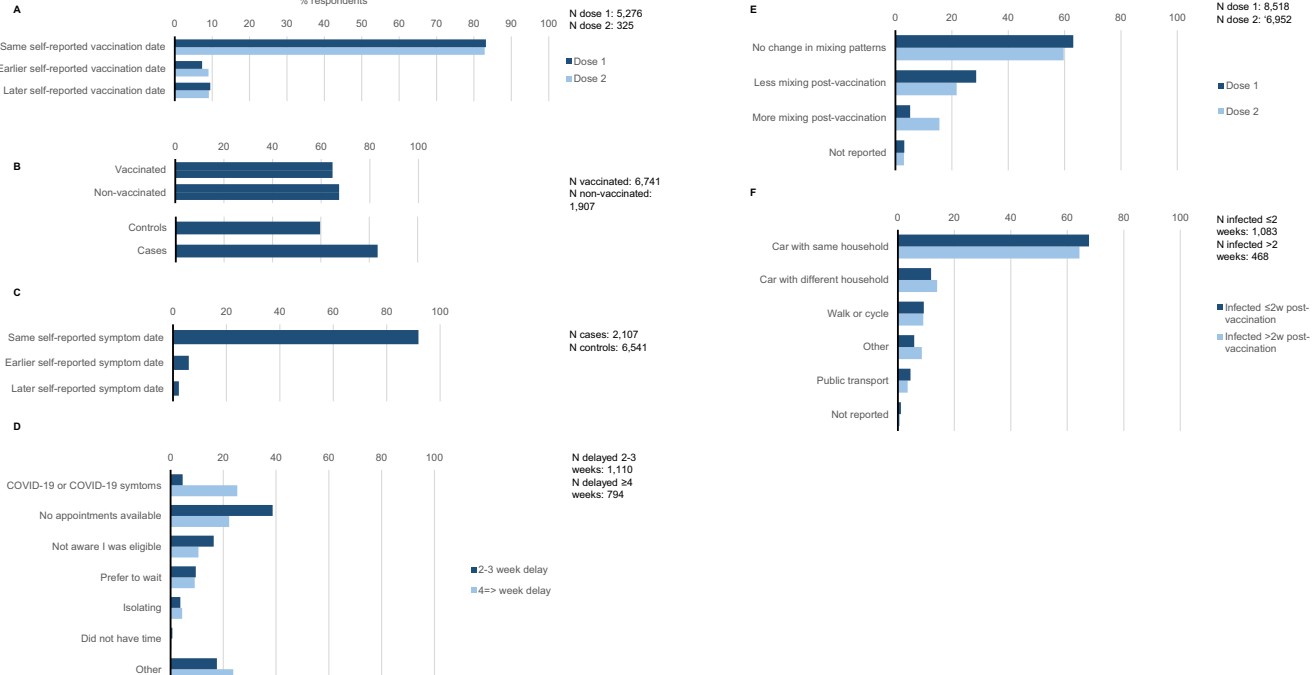

**Fig. 3 | Summary of results for assessment of bias and alternative causal pathways. A** Exposure (vaccination status) misclassification: comparison of vaccination dates in NIMS versus the questionnaire, by vaccination dose. **B** Outcome misclassification by symptomatic status: individuals reporting in the questionnaire they were symptomatic, by case and vaccination status. **C** Outcome misclassification by symptom onset date: comparison of onset dates in SGSS versus the questionnaire. **D** Deferral bias: individuals reporting they delayed their vaccination because they had COVID-19 or COVID-19 like symptoms, by length of vaccine delay from invitation. **E** Riskier behaviour after vaccination: individuals reporting they mixed more, the same or less after their vaccination, by vaccination dose. **F** Vaccination visit transit mode in relation to COVID-19: individuals reporting their mode of transport to the vaccination centre, amongst those with a positive COVID-19 test ≤2 weeks versus those with a positive COVID-19 test >2 weeks.

CI: 67–90%; Fig. 2). When assessing the vaccination status of those that deferred 2–3 or 4 weeks because of COVID-19/COVID-19 like symptoms we found 96% were non-vaccinated, 4.0% had one dose and 0.0% two doses.

When accounting for all of the above potential biases in the original TNCC study, vaccine effectiveness after two doses of BNT162b2 decreased slightly from 88% (95% CI: 79–94%) to 85% (95% CI: 68–94%; Fig. 2).

**Potential alternative causal pathways in original TNCC study**

When asking individuals in the questionnaire if they mixed more after their vaccination for assessment of alternative causal pathways via riskier behaviour after vaccination, 5.2% (445/8518) with a first dose and 15.6% (1087/6952) with a second dose, reported that they mixed more after their vaccinations, whereas the remaining 91.6% (7806/8518) for first dose and 81.4% (5658/6952) for second dose reported that they mixed the same or less (Fig. 3E and Table S2). Amongst those

that were vaccinated before symptom onset date, there was no association between mixing more and odds of COVID-19 when compared with those that reported mixing less or the same after any first dose vaccination (odds ratio [OR]: 0.92 95% CI: 0.68–1.24) after adjusting for age, gender, ethnicity, CEV, immunosuppressive conditions and month of vaccination dose. Due to a lack of individuals the same could not be assessed after second dose vaccinations.

When asking individuals in the questionnaire their mode of transport to vaccination centres, for assessment of alternative causal pathways for contracting COVID-19 individuals with a positive test within 2 weeks of vaccination did not appear to take risker types of transport compared to those that had a positive test after 2 weeks of vaccination (within 2 weeks: public transport: 4.5% [21/468], car with member outside of household: 11.8% [55/468]; after 2 weeks: 3.5% [38/1083] and 13.9% [151/1083], respectively; Fig. 3F and Table S2). Amongst those that were vaccinated before symptom onset date, there was no association between riskier transport to the vaccination centre and odds of COVID-19 (car with members outside household: OR: 1.28 95% CI: 0.98–1.67; public transport: OR: 1.26 95% CI: 0.81–2.03) when compared with those that took less risk forms of transport (drove alone or walked/cycled) after adjusting for age, gender, ethnicity, region and IMD. Therefore, there appeared to be no or minimal evidence of alternative causal pathways through riskier behaviour after vaccination or vaccination itself being associated with COVID-19 in the original TNCC study.

## Discussion

Among 23713 symptomatic individuals with a positive PCR test between 1 and 21 February 2021 in England, 8648 responded to a questionnaire to assess for potential bias in an influential TNCC study of COVID-19 vaccine effectiveness. Using information from the questionnaire to produce a combined estimate that accounted for all potential biases decreased the original vaccine effectiveness estimate after two doses of BNT162b2 from 88 to 85%. Self-reported behaviour demonstrated no or minimal evidence of riskier behaviour after vaccination.

The response rate to the questionnaire was lower in those with a negative PCR test, and also among people living in areas of greater deprivation or with non-White ethnicities, both associated with increased risk of COVID-19 related death[15]. However, there was a similar response rate by vaccination status, and vaccine effectiveness point estimates were very similar in the respondent versus non-respondent and overall questionnaire samples, suggesting the selection bias did not materially affect the vaccine effectiveness estimates.

Vaccination dates and symptom onset dates were consistent between the nationwide vaccination-COVID-19 PCR testing data (NIMS-SGSS) and the questionnaire with the majority of individuals self-reporting the same date as in NIMS-SGSS. When using self-reported vaccination dates, vaccine effectiveness decreased from the original estimate of 88 to 84%. When using self-reported onset dates vaccine effectiveness decreased marginally from 88 to 87%. For vaccinations other than for COVID-19 self-reported dates have previously been shown to be unreliable[16], however in the UK, individuals were asked to carry their COVID-19 vaccination cards[17] which could explain why self-reported vaccination dates were more reliable than expected. Vaccination status using self-reported dates was more likely to be different to vaccination status when using NIMS when age increased. This likely represents the greater impact of recall bias (i.e., the questionnaire was sent in March 2021 and individuals were still responding in August 2021 and it is likely that responses to this question became more unreliable with increasing number of days between the event occurring and response to the questionnaire) in older individuals[18]. For onset date, we likely underestimated misclassification since individuals were only asked to report their onset date in the questionnaire if different from the SGSS date that was provided. It is likely that some individuals

could not recall the date and left this field blank, which would have been inaccurately determined as the correct date, rather than missing.

Somewhat surprisingly, only 65.5% of individuals self-reported that they were symptomatic in the questionnaire, despite all having to be symptomatic at the time of requesting their PCR test. Cases were more likely to report being symptomatic in the questionnaire compared with negative controls, which resulted in a modest increase in vaccine effectiveness estimates (88 to 92%) when self-reportedly asymptomatic individuals were excluded. These findings may reflect a degree of outcome misclassification in the original study. They may also indicate a retrospective reassessment of symptom status by survey participants, including the downgrading of symptoms among individuals whose SARS-CoV-2 test was negative. Studies have previously found that specific comorbidities[15], household size and type[19,20] are highly associated with COVID-19. In the current study it was reassuring that when adjusting for individual CEV, household size and type, the vaccine effectiveness estimates decreased marginally from the original estimate of 88 to 87% providing limited evidence of confounding from these COVID-19 risk factors in the original TNCC study. When using these data to assess COVID-19 effectiveness early on in the pandemic, we can be more confident that missing information on these COVID-19 risk factors was less of a concern, although there could be confounding from other variables that were not collected in the questionnaire (e.g., mobility status). Factors such as occupation may also be key confounders in younger adults, but were assumed to be less relevant in the current study given our focus on individuals over 70 years of age.

Vaccine deferral because of COVID-19/COVID-19 symptoms was relatively common in the study. When we excluded individuals who deferred their vaccination because of COVID-19/COVID-19 like symptoms, vaccine effectiveness estimates decreased from the original estimate of 88 to 81%. A decrease in estimated effectiveness is expected given that this approach entails removing non-vaccinated individuals who received a positive COVID-19 test from the analysis. The effect of deferral bias appears to be modest and does not undermine conclusions from the original TNCC study regarding the high effectiveness of vaccines during the initial phases of implementation.

The combined vaccine effectiveness estimate that accounted for all potential biases saw a modest decrease in effectiveness from the original estimate of 88 to 85%. Although this small change is reassuring, 85% should not be considered a best estimate since questionnaire responses that were provided in some cases many months after the events occurred cannot be considered the gold standard.

The findings on riskier behaviour are interesting. Previously authors have suggested that information on individuals' risk behaviours and exposures should be collected when conducting vaccine effectiveness studies[13]. However, our study findings suggest that during the early stages of the pandemic in England in the elderly population, when the country was in lockdown there was low prevalence of risky behaviours following vaccination. Self-reported riskier behaviours might be susceptible to underreporting due to the impact of social desirability bias (wherein people are more likely to report behaviour in line with rules and recommendation). The lack of a significant association between mixing more after vaccination and the odds of COVID-19 may also reflect recruitment bias within the test-negative study population, whereby riskier behaviour increases exposure to both positive and negative (non-COVID-19) causes of symptoms. It would be beneficial to verify these analyses with other study designs. The widespread implementation of precautionary measures such as mask usage and physical distancing on public transport and vaccine centres may account for the lack of association between riskier transit types and COVID-19 risk.

A strength of the study was the large questionnaire sample size that meant the sample was fairly generalisable and allowed the identification of small differences in vaccine effectiveness estimates. This

study addressed an important evidence gap: previous literature[21–28] used theoretical proofs or simulated data to show the impact of substantial bias on different observational study designs. However, the current study used real-world data to detect the presence or absence of each of these biases and then quantified the true impact on vaccine effectiveness estimates. Another key strength of the study was that it assessed the robustness of an influential TNCC study that was one of the first observational studies that was used to inform governmental policy at the start of the pandemic[10].

However, despite these strengths, there were also a number of limitations. The study behavioural findings (e.g., mixing patterns) are likely only relevant to the period of time when the national population was under a strict lock down. Later on in the pandemic, when restrictions started to be lifted and individuals became "fatigued", it is likely that mixing patterns would be different to those identified in the current study[29]. Another limitation was that we were unable to assess whether collider bias[30–32] was present in the original study. Collider bias, another form of selection bias, could have potentially been introduced through the test-negative design. This type of bias is potentially introduced as health-seeking behaviour is associated with testing, vaccination uptake and infection i.e., testing is a 'collider' on the pathway between vaccination and infection[30,31,33]. We could not determine the presence of this bias because the association between health-seeking behaviour and testing could not be assessed as this information is not recorded in the data. Future studies should collect information on health-seeking behaviour so that this association can be assessed.

Based on the findings from the current study, policy makers can be more confident in their decisions made and other policy decisions that were made using the same study design in this population early on in the pandemic. Similarly, this study provides some reassurance on ongoing national vaccine effectiveness studies using TNCC with the same data sources, to support the public and healthcare workers to have continued confidence in reports of vaccine effectiveness e.g., against new strains. Future studies are required to determine whether the current findings remain applicable now that restrictions have been lifted.

Overall, there appeared to be minimal evidence of any large biases that may have affected an important TNCC COVID-19 vaccine effectiveness study that informed governmental policy early in the COVID-19 pandemic. Based on this, clinicians and policy makers can be more confident in any decisions that were made based on this study and in the TNCC studies that were conducted throughout the pandemic to assess vaccine effectiveness against new variants and to assess duration of protection of the vaccinations.

## Methods
### Data sources
The original study used national vaccination (NIMS) and COVID-19 testing in the community (pillar 2; SGSS) data, linked at the patient level. Details on the variables available at the time of the original analysis can be found in Table S5.

The new questionnaire data was used in combination with the NIMS and SGSS data used in the original study[4]. Data from these sources were linked on the patient level. The questionnaire was sent in March 2021 to the subset of individuals from the original TNCC study[4] who had a PCR COVID-19 test between 1 February 2021 and 21 February 2021. The most recently tested individuals were selected in order to minimise the impact of recall bias. The questionnaire (Materials S1) aimed to collect the necessary information required to assess for the presence of specific biases and behavioural changes related to vaccination. This included COVID-19 vaccination dates, COVID-19 risk factors (including comorbidities that would qualify an individual as high risk for COVID-19[11], CEV status[34], care home status, household size and household type), time from vaccination invitation to vaccination (first

and second dose), reasons for vaccination delay if vaccinated more than two weeks after invitation, reasons for no vaccination if not vaccinated, social mixing behaviours after vaccination (first and second dose), mode of transportation to vaccination centres (first and second dose), COVID-19 onset dates and symptomatic status. The COVID-19 testing and symptom date of interest were specified in the survey letter (Materials S1).

### Study analyses
**Population description and selection bias.** To assess whether the questionnaire responses were representative of all ≥70 year olds in England that had their first PCR test in February 2021 the demographics and clinical characteristics (age, gender, ethnicity, geographical region, IMD, week of COVID-19 symptom onset, week of COVID-19 test, care home status, test result, CEV status and COVID-19 vaccination status) of respondents of the questionnaire were described on 31 March 2021 and compared with non-respondents using percentage difference (with +/−5% absolute difference as threshold to define clinically meaningful differences) and Chi-squared/Fisher's exact test. Any missing data was described.

To assess whether potential selection bias had been introduced through the questionnaire sampling or response, the original vaccine effectiveness estimates (i.e., the odds of vaccination in cases and negative controls estimated using logistic regression models adjusted for potential confounders that were available in the data at the time: age, gender, ethnicity, geography, index of multiple deprivation (IMD), care home status and week of onset) were run on the entire questionnaire sample and then amongst those that responded ("respondents") and did not respond ("non-respondents"). As in the original study vaccine effectiveness was estimated as (1 − odds ratio) x 100.

Respondents that were vaccinated were compared to non-vaccinated and cases were compared to negative controls based on demographics and clinical characteristics. These were compared using percentage difference (with +/−5% absolute difference as a threshold to define clinically meaningful differences) and Chi-squared/Fisher's exact test and missing data was also described.

Sources of potential bias and behavioural changes related to vaccination were then explored among questionnaire respondents as outlined below.

**Potential biases in original TNCC study.** Vaccination status could be misclassified in NIMS if vaccination dates are incorrect (Fig. S1). The questionnaire, therefore, asked participants to self-report their vaccination date to identify any exposure misclassification. The number of individuals with the same, earlier or later self-reported vaccination date compared with NIMS was described as well as the distribution in difference in days using histograms for both doses. We also described the number of self-reported vaccination dates that were within 3 days +/− of NIMS (inclusive) or more and less than 3 +/− days for both doses.

We updated vaccination status using self-reported vaccination date, and if this field was missing in the questionnaire, we used the NIMS date. Amongst this population, we reported vaccination status based on self-reported vaccination dates and to assess for the potential impact of exposure misclassification on vaccine effectiveness estimates, we ran the logistic regression models from the original study (see above) using self-reported vaccine dates. To explore the potential mismeasurement of exposure misclassification within levels of confounders we described key confounders (age, gender, ethnicity, geography, index of multiple deprivation (IMD), week of onset, care home status and CEV) amongst those identified with increased or decreased number of vaccine dose counts when using self-reported vaccine dates (versus NIMS) compared to those with no change in vaccine status. These were compared using percentage difference (with +/−5% absolute difference set as threshold) and Chi-squared/Fisher's exact test.

The symptomatic status could be misclassified in SGSS if individuals incorrectly report they are symptomatic at the time of requesting their PCR test either to access free testing or because they are concerned about mild/vague symptoms (Fig. S1). This could also have affected the selection of the study population, since only symptomatic individuals were eligible for inclusion. Therefore, to assess for potential outcome misclassification through symptomatic status, the self-reported symptomatic status in the questionnaire was compared to the status reported in SGSS. Since all individuals identified in SGSS reported that they were symptomatic, the proportion of this population that reported they were asymptomatic in the questionnaire overall and by case and vaccination status was reported. The denominator in this population was all those responding to the symptomatic status question in the questionnaire. The logistic regression models from the original study (see above) were re-run amongst the population of individuals that reported they were symptomatic in the questionnaire. To explore the potential mismeasurement of outcome misclassification within levels of confounders we described key confounders (as above) amongst those self-reporting asymptomatic versus symptomatic status. These were compared using percentage difference (with +/−5% absolute difference set as threshold) and Chi-squared/Fisher's exact test.

The onset date could be misclassified in SGSS if individuals incorrectly reported their symptom onset date when booking their PCR test (Fig. S1). Individuals that reported they were symptomatic in the questionnaire were asked to report their symptom onset date (if different from the date in SGSS which was provided in the questionnaire) to assess for systematic differences. The number of individuals with the same, earlier or later self-reported onset date compared with SGSS was described as well as the distribution in difference in days using a histogram. We also described the number of self-reported onset dates that were within 3 days +/− of SGSS (inclusive) or more and less than 3 +/− days.

Vaccination status using self-reported symptom onset date from the questionnaire was updated and amongst this population, we reported vaccination status and ran the logistic regression models from the original study (see above). However, this would be interpreted with caution a priori because of the potential impact of recall bias[35]. To explore the potential mismeasurement of outcome misclassification within levels of confounders we described key confounders (as above) amongst those self-reporting different versus same onset date in the questionnaire. These were compared using percentage difference (with +/−5% absolute difference set as threshold) and Chi-squared/Fisher's exact test.

Confounding from COVID-19 risk factors was potentially present in the original study since it was not possible at the time to identify comorbidities and other risk factors for COVID-19 (e.g., household size and type) using NIMS and SGSS (composite variables including any risk group and CEV, have since been added but individual conditions remain unavailable in these datasets; Fig. S1). Therefore, to assess for potential confounding, the logistic regression models from the original study (see above) were repeated additionally adjusting for each potential COVID-19 risk factor in turn obtained from the questionnaire, including: CEV; the number of persons per household; household type; immunosuppression (separately and combined: HIV/immunodeficiency, organ or bone marrow transplant, immunosuppression due to medication and asplenia or dysfunction of the spleen); and other comorbidities that qualify an individual as high risk (separately and then combined: chronic heart disease, chronic kidney disease, chronic respiratory disease excluding asthma, cancer, seizure disorder, chronic liver disease, asthma requiring medication, chronic neurological disease and BMI ≥ 40 kg/m$^2$). The pre-specified analysis plan was to include all variables which changed the odds ratio of vaccination by 0.01 amongst the PCR-confirmed individuals in a multivariable model.

Deferral bias[36–38] is potentially introduced if individuals delay their vaccinations because they have a COVID-19 infection, COVID-19 like symptoms or have been recently exposed to COVID-19 (individuals in the UK are asked to delay their vaccine by 28 days if they contract COVID-19[14]; Fig. S1). Individuals that decide to defer their vaccination because of this might then go on to test positive for COVID-19 which leads to a temporary apparent protective effect of the vaccination in recently vaccinated individuals[36]. Therefore, to assess for potential deferral bias the proportion of individuals who reported they received their vaccinations ≥4 weeks from their invitation because they had COVID-19 or COVID-19 like symptoms was reported and the proportion of individuals that reported they had not yet been vaccinated because they had been unwell or had COVID-19 was also reported. The denominator population was all individuals reporting they were ever vaccinated with a first dose or second dose in the questionnaire. To assess by how much deferral bias might be expected to increase vaccine effectiveness estimates, we ran the logistic regression models from the original study (see above) removing individuals that reported they delayed either 2–3 weeks or 4 weeks because of COVID-19/COVID-19 like symptoms. We also described the vaccination status at symptom onset date of those that deferred their vaccination 2–3 or 4 week because of COVID-19/COVID-19 like symptoms.

When accounting for all biases at once, we ran the logistic regression models from the original study (see above) amongst those that did not delay their vaccination because of COVID-19/COVID-19 like symptoms, that self-reported they were symptomatic and using vaccination and symptom onset dates from the questionnaire adjusting for CEV, household size and type (as well as confounders adjusted for in the original TND study; Fig. 2).

**Potential alternative causal pathways in original TNCC study.** If vaccinated individuals start mixing more with individuals outside of their household after being vaccinated, then the risk of contracting COVID-19 might increase in these individuals creating an "alternative causal pathway" from vaccination to infection (Fig. S1). If increased mixing occurs at a faster rate compared to non-vaccinated individuals' then this could lower vaccine effectiveness estimates compared to true estimates. To assess for riskier behaviour after vaccination the proportion of those that reported that they mixed the same, more or less in the 3–4 weeks after the date of their first or second vaccination was reported. Amongst those that were vaccinated before the symptom onset date, the odds of COVID-19 amongst those that reported they mixed more were compared to those that mixed less or the same, using logistic regression adjusting for potential confounders (age, gender, ethnicity, CEV, immunosuppressive conditions and month of vaccination dose).

Other alternative causal pathways are potentially introduced if individuals' contract COVID-19 on the way to or back from their vaccination centres (Fig. S1). These pathways include a composite of events immediately before and after vaccination, though in practice all exposures would precede the induction of robust vaccine immunity. Exposures at the time of vaccination could have potentially lowered vaccine effectiveness estimates compared to true vaccine effectiveness estimates, especially early on in the pandemic when individuals were instructed to stay at home if they were not carrying out certain tasks (e.g., going to get vaccinated, food shopping etc.). To assess for travel to the vaccination itself being associated with COVID-19 the mode of transport to and from vaccination centres (first and second dose) was reported amongst those that had a positive COVID-19 test within 2 weeks of vaccination, compared with those that had a positive test after 2 weeks. Amongst those who were vaccinated before symptom onset date, the odds of COVID-19 amongst those who travelled to and from their vaccination centre in a car with someone outside of their household or on public transport (i.e., riskier transport modes) was compared to those that travelled either alone in a car or walked/cycled (i.e., less risky transport modes) using logistic regression

adjusted for age, gender, ethnicity, region and IMD, since these variables were likely to be associated with mode of transportation and COVID-19 risk.

All of the analyses were conducted using Stata (version 17) and R (version 4.1.3).

**Ethics.** This analysis was conducted as part of public health service evaluation. UKHSA has legal permission, provided by Regulation 3 of The Health Service (Control of Patient Information) Regulations 2002 to process patient confidential information for national surveillance of communicable diseases and as such, individual patient consent is not required to access records. Research ethics approval was therefore not sought.

### Reporting summary
Further information on research design is available in the Nature Portfolio Reporting Summary linked to this article.

## Data availability
Access to pseudonymised national datasets used in this study (National Immunisation Management Service and Second Generation Surveillance System) is managed by NHS England through the NHS COVID-19 Data Store: https://www.england.nhs.uk/contact-us/privacy-notice/how-we-use-your-information/covid-19-response/nhs-covid-19-data-store/. Questionnaire data was collected for the purposes of public health service evaluation and consent was not obtained for further sharing for research. To discuss a request for UKHSA data you would like to submit, contact DataAccess@ukhsa.gov.uk.

## Code availability
The programming code for this project is available on Github: https://github.com/grahams99/Enhanced_surveillance_questionnaire.

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

## Acknowledgements

This work uses data provided by patients and collected by the NHS as part of their care and support (http://www.usemydata.org). In addition, we would like to thank participants of the questionnaire who provided valuable data that enabled the conduct of this study. S.G., E.M., N.A., J.L.W. and N.A. and H.I.M. are funded by the National Institute for Health and Care Research (NIHR) Health Protection Research Unit in Vaccines and Immunisation (grant reference NIHR200929), a partnership between UK Health Security Agency and London School of Hygiene & Tropical Medicine. EPKP received funding from the UKRI COVID-19 Longitudinal Health and Wellbeing National Core Study (Phase 1 LHW-NCS, MC_PC-20059). The views expressed are those of the author(s) and not necessarily those of the NIHR, UK Health Security Agency or the Department of Health and Social Care.

## Author contributions

Study concepts, E.T., J.S., J.L.B., E.M., N.A.; Study design, S.G., E.T., J.S., J.L.B., D.N., E.M., N.A., J.L.W., H.I.M.; Data acquisition, E.T., J.S., J.L.B., E.M., N.A.; Programming, S.G, E.T., J.S.; Statistical analysis, S.G.; Supervision, N.A., J.L.W., H.I.M., E.P.; Analysis and interpretation of results, All authors; Manuscript preparation, S.G.; Manuscript editing, E.P., D.N., E.T., N.A., J.L.W., H.I.M.; Manuscript review, All authors; Manuscript approval, All authors.

## Competing interests

The UK Health Security Agency (UKHSA) has provided vaccine manufacturers with post-marketing surveillance reports which the companies are required to submit to the UK Licensing Authority in compliance with their Risk Management Strategy, and a cost recovery charge is made for these reports. SG is also a part-time salaried employee of Evidera, which is a business unit of Pharmaceutical Product Development (PPD), part of Thermo Fisher Scientific. The remaining authors declare no competing interests.
