## [Peer Review File · Nature Communications]

Bias assessment of a test-negative design study of COVID-19 vaccine effectiveness used in national policymakingREVIEWER COMMENTS

Reviewer #1 (Remarks to the Author):

In this article, the investigators behind the test negative studies conducted in the UK describe the results of a survey questionnaire sent out to a subset of individuals tested in February 2021. The purpose was to examine whether additional covariates that were not available in routinely collected data (e.g. household size, household type, and clinical vulnerability) were confounders. The other purpose was to assess how changes to risk behavior around the time of vaccination or after vaccination can affect the results.

Overall, this is a very useful study. Though some of the behavioral results may not generalize to the current time period (as noted by the authors), it is an important exercise in testing the limits of their original study, and it provides a blueprint for how future studies might do the same. In fact, what they find is pretty reassuring. Clinical vulnerability and household size/type are not observed to be confounders. Even analyzing the results just among survey respondents returns estimates similar to the overall findings. The study examines the quality of self-reported/recalled data, like vaccination timing and symptom status. This is very interesting raw data as well. The high percent reporting themselves as asymptomatic is intriguing. I don't know that the authors had a great explanation for that – certainly worth further examination. And then they were able to collect data on reasons for vaccine deferral, etc.

I have one comment, and that is with regards to the risk behavior analysis. It seems like the investigators are fitting a logistic regression model to the test negative data to return an odds ratio comparing those reporting more vs. less/same risk behavior. But wouldn't more risk behavior have potential to increase both COVID-19 AND non-COVID causes of symptoms? The only types of associations that can be meaningfully estimated in test negative studies are for interventions that are highly specific (like vaccination). But general "risky behavior" could increase positive and negative causes of symptoms alike, leading to a null finding. So, it would be helpful for the authors to explain how this analysis tests for this "alternative causal pathway." That was not clear.

Minor:

- Methods section on "Riskier behavior after vaccination" has a typo. "Odds of COVID amongst those that reported they mixed more was compared to those that mixed more [less] or the same."

Natalie Dean
Emory University

Reviewer #2 (Remarks to the Author):

This is an interesting approach to estimating the degree to which VE estimates from the UK have been biased. However, it feels like it is only half done. The authors make some sweeping claims about not bothering to use the survey sample to actually assess bias in the data because the differences look small and therefore wouldn't be a problem. This is counter to so much of epidemiologic methods literature, which has shown that even small differences, when differential and/or dependent, can drastically harm effect estimates. The paper therefore feels like it is only half done.

1. Lines 94-107: The authors state that sample was different from the population in numerous ways. This assessment of difference is presumably based on p-values. The sample was huge, so very minor differences are shown to be statistically significant, but it is dubious whether these are meaningful. It might be more helpful to set a threshold of difference that is concerning. E.g. a 5 percentage point or a fold difference. Of all the effects listed the only differences that seem concerning (IMHO) are the test results (nearly 10 percentage points difference) and then maybe age, deprivation and ethnicity, but it is subjective.

2. Suppl Table S3. It would be easier to eyeball the similarities/differences if the table was organised differently. What we are interested in comparing is the responders and non-responders. So, either a wide forest plot organised in landscape with the questionnaire sample, responders, non-respondents side-by-side, or restructure the table to show the different samples side-by-side.

And drop the p-values – they are unnecessary. Nature group has previously published on the futility of p-values: <https://www.nature.com/articles/d41586-019-00874-8>
<https://www.nature.com/articles/d41586-019-00857-9>

3. Exposure misclassification, lines 115-122: the authors state “Therefore, there appeared to be no or minimal evidence of systematically inaccurate dates in NIMS when compared with self-reported vaccination date”. This statement needs to be supported by further analysis. The differences in recall matter with respect to case status and key confounders and they should not be expected to just cancel each other out. What is this information used for? Waning analysis? It would be more helpful if the authors conducted a sensitivity analysis of their original data reclassifying the misclassified individuals and then re-estimating VE. Simple methods to do this using a 2x2 table are available, e.g. <https://pubmed.ncbi.nlm.nih.gov/9027513/> and there are Stata and R packages available. More details discussion of what to do without validation data is available, and the authors can leverage the advantages of their study accordingly:
<https://doi.org/10.1093/ije/dyx027>.

4. The DAG does not show any measurement errors, usually depicted with an asterisk. It would be helpful to depict the bias explored in each section with reference to the associations shown in the DAG. It may well be the case that more than one DAG is needed for brevity in explanation (as well as one that shows all the associations).

5. The authors have a unique opportunity to also assess whether this misclassification was dependent on any confounders so that they can also explore the impact of dependent differential exposure misclassification. This could be explored using deterministic or probabilistic bias analysis or Bayesian methods. The DAG does not reflect any dependencies in classification and identification of them using this survey sample would be really helpful.

6. Outcome misclassification, lines 124-134: This isn't outcome misclassification; the presence of symptoms for both cases and controls was an eligibility criterion in a test negative design and therefore this section is discussing selection bias. The re-estimation of VE is in the truly eligible sample and therefore VE against symptomatic disease, instead of whatever it was estimated against before, which sounds like a mixture of symptomatic and asymptomatic disease. It would be helpful if the mismeasurement of eligibility was explored within levels of confounders as well as outcome/exposure. As predicted by Lewnard et al. this has led to bias away from the null. <https://pubmed.ncbi.nlm.nih.gov/34001753/>. Other considerations about the action of this bias in case-control samples were discussed by Jurek et al. <https://doi.org/10.1016/j.annepidem.2012.12.007>.

7. Because this is selection bias it would be more logical to list this first – before exposure misclassification. If doing a probabilistic bias analysis, the biases are reconstructed in the reverse order in which they occurred, so thoughtful consideration as to the order of bias is important.

8. Lines 133-4: the cross-reference for the relevant table or figure hasn't worked.

9. Outcome misclassification, lines 136-146: Again this section is mislabelled. What is symptom onset date used for? It can be used as a restriction criterion for recruitment (so is a sampling/selection issue), or it can be used to measure the exposure if estimating waning VE (so is an exposure measurement error). Both impacts should be explored in the data. It is not acceptable to assume that there will be minimal difference. It has been shown that small differences can impact effect estimates. E.g. <https://doi.org/10.1093/ije/dym291> (other examples exist).

10. Confounding, lines 148-157: one of the key confounders in COVID-19 VE studies has been eligibility for vaccination and this highly conflated with occupation. E.g. in most countries, healthcare workers were prioritised for vaccination because they were considered to be at higher risk of infection. However, I don't see this in the DAG or the list of confounders explored in the survey. Why not?

11. Confounders are all grouped together in the DAG, but this is unrealistic because they can be dependent on other nodes and the confounding caused may be direct or via other nodes to create backdoor paths. The DAG may need to be exploded to show each (or just a selection) confounder's many paths.

12. Healthy vaccinee bias, lines 159-172: For clarity, the authors should state up front what kind of bias they think this is. I don't see it depicted in the DAG. It reads like this is a source of confounding bias and effect modification; it could also be causing some collider bias. Again, the authors have claimed that this is unlikely to affect their estimates, but it would not be too hard to actually check this in the data.

13. Riskier behaviour, lines 174-184: again, is this confounding bias? It reads like it is causing

confounding and effect modification. In Figure 1 it is shown as an intermediate. So, why not look at its effect by doing a very simple mediation analysis to determine the direct and indirect effects?

14. Vaccination transportation, lines 186-198: this is depicted as a mediator in the DAG in Figure S1. However, the description makes it sound more like a selection bias issue. Is this really a form of collider bias? The Methods don't describe the mechanism clearly. Because DAGs need to show temporality, transport to the vaccination centre precedes the exposure so should be an antecedent of vaccination, whereas travelling from is descendent, so there probably needs to be more than one node for transport in the DAG.

15. It would be really nice if the authors took all these forms of bias and did a probabilistic (or deterministic) sensitivity analysis. Modern Epidemiology 4 has a nice primer on this (chapter on bias analysis) and there is also one in "Applying Quantitative Bias Analysis to Epidemiologic Data" by Tim Lash and co. There is a package in Stata by Nicola Orsini that makes implementation a little more straightforward

(<https://journals.sagepub.com/doi/pdf/10.1177/1536867X0800800103>). There is probably also one for R. Of course, this needs to also acknowledge limitations of bias analysis, e.g.

<https://doi.org/10.1093/aje/kwab069>.

16. The conclusions about the degree to which certain forms of bias were important or not would be easier to accept if the additional suggested analyses were performed to confirm the authors' suspicions.

17. The Discussion does not address the sampling bias that may be present in this study. There have been several papers discussing the utility of validation samples. I don't see any of these cited in the discussion. E.g. <https://doi.org/10.1093/ije/dyy138>, [https://doi.org/10.1016/0895-4356\(88\)90020-0](https://doi.org/10.1016/0895-4356(88)90020-0),

18. In the Methods, can it be clarified whether respondents were reminded of the period that was relevant to them and asked to only think about the behaviours in that period?

19. I didn't see anywhere the statistical program that was used for analysis. Was it R? Stata? SAS? Something else?

Apologies if I missed this detail.

REVIEWER COMMENTS

Reviewer #1 (Remarks to the Author):

In this article, the investigators behind the test negative studies conducted in the UK describe the results of a survey questionnaire sent out to a subset of individuals tested in February 2021. The purpose was to examine whether additional covariates that were not available in routinely collected data (e.g. household size, household type, and clinical vulnerability) were confounders. The other purpose was to assess how changes to risk behavior around the time of vaccination or after vaccination can affect the results.

Overall, this is a very useful study. Though some of the behavioral results may not generalize to the current time period (as noted by the authors), it is an important exercise in testing the limits of their original study, and it provides a blueprint for how future studies might do the same. In fact, what they find is pretty reassuring. Clinical vulnerability and household size/type are not observed to be confounders. Even analyzing the results just among survey respondents returns estimates similar to the overall findings. The study examines the quality of self-reported/recalled data, like vaccination timing and symptom status. This is very interesting raw data as well. The high percent reporting themselves as asymptomatic is intriguing. I don't know that the authors had a great explanation for that – certainly worth further examination. And then they were able to collect data on reasons for vaccine deferral, etc.

Response: Thank you for the positive feedback. We are pleased that you found the study useful. We agree that the high number of participants reporting that they were asymptomatic is interesting and merits further consideration. We have added the following paragraph to the discussion to address this:

“Somewhat surprisingly, only 65.5% of individuals self-reported that they were symptomatic in the questionnaire, despite all having to be symptomatic at the time of requesting their PCR test. Cases were more likely to report being symptomatic in the questionnaire compared with negative controls, which resulted in a modest increase in vaccine effectiveness estimates (88% to 92%, albeit with overlapping confidence intervals) when self-reportedly asymptomatic individuals were excluded. These findings may reflect a degree of outcome misclassification in the original study. They may also indicate a retrospective reassessment of symptom status by survey participants, including the downgrading of symptoms among individuals whose SARS-CoV-2 test was negative.”

I have one comment, and that is with regards to the risk behavior analysis. It seems like the investigators are fitting a logistic regression model to the test negative data to return an odds ratio comparing those reporting more vs. less/same risk behavior. But wouldn't more risk behavior have potential to increase both COVID-19 AND non-COVID causes of symptoms? The only types of associations that can be meaningfully estimated in test negative studies are for interventions that are highly specific (like vaccination). But general “risky behavior” could increase positive and negative causes of symptoms alike, leading to a null finding. So, it would be helpful for the authors to explain how this analysis tests for this “alternative causal pathway.” That was not clear.

Response: We agree with your comment and have added the following discussion of this limitation: *“The lack of a significant association between mixing more after vaccination and the odds of COVID-19 may also reflect recruitment bias within the test-negative study population, whereby riskier behaviour increases exposure to both positive and negative (non-COVID-19) causes of symptoms. It would be beneficial to verify these analyses with other study designs.”*

Minor:

- Methods section on “Riskier behavior after vaccination” has a typo. “Odds of COVID amongst those that reported they mixed more was compared to those that mixed more [less] or the same.”

Response: Thank you for bringing this to our attention. We have now corrected this (replacing ‘more’ with ‘less’).

Natalie Dean
Emory University

Reviewer #2 (Remarks to the Author):

This is an interesting approach to estimating the degree to which VE estimates from the UK have been biased. However, it feels like it is only half done. The authors make some sweeping claims about not bothering to use the survey sample to actually assess bias in the data because the differences look small and therefore wouldn't be a problem. This is counter to so much of epidemiologic methods literature, which has shown that even small differences, when differential and/or dependent, can drastically harm effect estimates. The paper therefore feels like it is only half done.

Response: We are glad you found this to be an interesting approach, and are grateful for the detailed and constructive comments. We agree that a more systematic presentation of vaccine effectiveness accounting for each source of bias would be beneficial. As outlined in our responses below, we have added more detail throughout our analyses, re-estimating vaccine effectiveness for each individual source of bias based on self-reported survey results. We also provide a combined estimate of vaccine effectiveness accounting for all measured sources of bias from the survey. We feel that these changes have helped improve the rigour and completeness of the revised manuscript, and thank the reviewer for the valuable feedback.

9. Lines 94-107: The authors state that sample was different from the population in numerous ways. This assessment of difference is presumably based on p-values. The sample was huge, so very minor differences are shown to be statistically significant, but it is dubious whether these are meaningful. It might be more helpful to set a threshold of difference that is concerning. E.g. a 5 percentage point or a fold difference. Of all the effects listed the only differences that seem concerning (IMHO) are the test results (nearly 10 percentage points difference) and then maybe age, deprivation and ethnicity, but it is subjective.

Response: Thank you for this comment. We agree with the importance of focusing on clinically meaningful differences. As suggested, we have added columns to Table 1 and Supplementary Table 2 providing the percentage differences between groups. We have also edited the Methods and Results to highlight characteristics with a difference of at least 5 percentage points and p values of <0.05. Specific changes are as follows:

Methods:

- *“... and compared with non-respondents using percentage difference (with +/-5% set as threshold to define clinically meaningful differences) and Chi-squared/Fisher's exact test.”*
- *“Respondents that were vaccinated were compared to non-vaccinated and cases were compared to negative controls based on demographics and clinical characteristics. These were compared using percentage difference (with +/-5% as a threshold to define clinically meaningful differences) and Chi-squared/Fisher's exact test and missing data was also described.”*

Results:

- *“...with respondent being younger, more likely to be of White ethnicity, less likely to live in a deprived area, and more likely to be a case when compared with non-respondents (based on a percentage difference of +/-5% and p values of <0.05; Table S2).”*
- *“Respondents that were cases were more likely to be non-vaccinated, more likely to be from the Northeast and Yorkshire and less likely to be from the Southwest of England, more likely to be deprived and were more likely to have earlier testing compared with respondents that were negative controls (based on +/-5% percentage difference and p values of <0.05; Table 1).”*

2. Suppl Table S3. It would be easier to eyeball the similarities/differences if the table was organised differently. What we are interested in comparing is the responders and non-responders. So, either a wide forest plot organised in landscape with the questionnaire sample, respondents, non-respondents side-by-side, or restructure the table to show the different samples side-by-side. And drop the p-values – they are unnecessary. Nature group has previously published on the futility of p-values: <https://www.nature.com/articles/d41586-019-00874-8>
<https://www.nature.com/articles/d41586-019-00857-9>

Response: As suggested, we have re-organised the table so that the results are side by side and we have removed the p-values.

3. Exposure misclassification, lines 115-122: the authors state “Therefore, there appeared to be no or minimal evidence of systematically inaccurate dates in NIMS when compared with self-reported vaccination date”. This statement needs to be supported by further analysis. The differences in recall matter with respect to case status

and key confounders and they should not be expected to just cancel each other out. What is this information used for? Waning analysis? It would be more helpful if the authors conducted a sensitivity analysis of their original data reclassifying the misclassified individuals and then re-estimating VE. Simple methods to do this using a 2x2 table are available, e.g. <https://pubmed.ncbi.nlm.nih.gov/9027513/> and there are Stata and R packages available. More details discussion of what to do without validation data is available, and the authors can leverage the advantages of their study accordingly: <https://doi.org/10.1093/ije/dyx027>.

Response: Thank you for this comment. As suggested, we have extended the Results section to provide a more detailed analysis of the extent of the differences between self-reported vaccination date and NIMS records, as summarised via histograms in a new Supplementary Figure 2. We have also added the sensitivity analysis suggested by the reviewer, re-estimating vaccine effectiveness based on self-reported vaccination dates (as presented in a new row in Figure 2). The relevant changes that have been added into the manuscript are as follows:

Methods:

- *“The number of individuals with the same, earlier or later self-reported vaccination date compared with NIMS was described as well as the distribution in difference in days using histograms for both doses. We also described the number of self-reported vaccination dates that were within 3 days +/- of NIMS (inclusive) or more and less than 3+/- days for both doses.”*
- *“We updated vaccination status using self-reported vaccination date, and if this field was missing in the questionnaire, we used the NIMS date. Amongst this population we reported vaccination status based on self-reported vaccination dates and to assess for the potential impact of exposure misclassification on vaccine effectiveness estimates, we ran the logistic regression models from the original study (see above) using self-reported vaccine dates. To explore the potential mismeasurement of exposure misclassification within levels of confounders we described key confounders (age, gender, ethnicity, geography, index of multiple deprivation (IMD), week of onset, care home status and CEV) amongst those identified with increased or decreased number of vaccine dos counts when using self-reported vaccine dates (versus NIMS) compared to those with no change in vaccine status. These were compared using percentage difference (with +/-5% set as threshold) and Chi-squared/Fisher’s exact test.”*

Results:

- *“89.8% of first doses self-reported in the questionnaire were within 3 days +/- of NIMS date (inclusive), whereas 3.6% were more than 3 days earlier and 6.6% were more than 3 days later in the questionnaire. For second dose, 93.3% of self-reported vaccination dates were within 3 days +/- NIMS date, with 1.0% were 3 days earlier and 5.8% were more than 3 days later (Figure S2A and S2B).”*
- *“When updating vaccination dates to those self-reported in the questionnaire (or if missing using vaccination dates from NIMS), the percentage of individuals identified as non-vaccinated at symptom onset date (using SGSS) increased very slightly from 22.1% when using NIMS to 23.5% when using the questionnaire. Vaccine effectiveness after two doses of BNT162b2 decreased marginally from 88% (95% CI: 79-94%) to 84% (95% CI: 74-92%; Figure 2).”*

Discussion:

- *“When using self-reported vaccination dates, vaccine effectiveness decreased from the original estimate of 88% to 84% (with overlapping confidence intervals). When using self-reported onset dates vaccine effectiveness decreased marginally from 88% to 87% (with overlapping confidence intervals). For vaccinations other than for COVID-19, self-reported dates have previously been shown to be unreliable¹⁶, however in the UK, individuals were asked to carry their COVID-19 vaccination cards¹⁷ which could explain why self-reported vaccination dates were more reliable than expected. Vaccination status using self-reported dates was more likely to be different to vaccination status when using NIMS when age increased. This likely represents the greater impact of recall bias (i.e., the questionnaire was sent in March 2021 and individuals were still responding in August 2021 and it is likely that responses to this question became more unreliable with increasing number of days between the event occurring and response to the questionnaire) in older individuals¹⁸.”*

4. The DAG does not show any measurement errors, usually depicted with an asterisk. It would be helpful to depict the bias explored in each section with reference to the associations shown in the DAG. It may well be the case that more than one DAG is needed for brevity in explanation (as well as one that shows all the associations).

Response: Thank you for your helpful feedback, we have provided a more comprehensive illustration of the pathways being explored in our study. We have therefore updated Supplementary Figure 1 to include all pathways of interest in this study. We have provided asterisks in the Figure where there is measurement error, as suggested. We believe this to be a clearer representation of our analysis framework than a full DAG, and we title the figure accordingly (“Key pathways under investigation in the current study”), with the figure legend: *“It should be noted that not all possible pathways are represented in the below figure, however, the key pathways are represented for exposure misclassification, outcome misclassification, confounding, deferral bias, riskier behaviour after vaccination and vaccination itself associated with COVID-19.”*

5. The authors have a unique opportunity to also assess whether this misclassification was dependent on any confounders so that they can also explore the impact of dependent differential exposure misclassification. This could be explored using deterministic or probabilistic bias analysis or Bayesian methods. The DAG does not reflect any dependencies in classification and identification of them using this survey sample would be really helpful.

Response: Thank you for this comment. As noted above, we have updated Supplementary Figure 1 to include all key pathways under investigation in the current study. As suggested by the reviewer, we have added a new Supplementary Table 5, which explores the association between key confounders and exposure misclassification (unchanged vs increased self-reported dose count and unchanged vs decreased self-reported dose count). We describe these in the Results as follows:

“When exploring key confounders amongst those with different self-reported vaccination status at symptom onset (from SGSS) versus unchanged status, we found that increased self-reported dose counts were associated with aged 75-79 years, most deprived IMD quintile, and COVID-19 symptom onset date in February week 1, but less associated with age 70-74 years (based on +/-5% percentage difference and p values of <0.05; Table S5). On the other hand, decreased self-reported dose counts were associated with male gender, COVID-19 symptoms testing in February week 2 and were less associated with age 70-74 years, the 4th quintile of deprivation (with 5th being the lowest), and COVID-19 symptoms in February week 3 (based on +/-5% percentage difference and p values of <0.05; Table S5).”

6. Outcome misclassification, lines 124-134: This isn't outcome misclassification; the presence of symptoms for both cases and controls was an eligibility criterion in a test negative design and therefore this section is discussing selection bias. The re-estimation of VE is in the truly eligible sample and therefore VE against symptomatic disease, instead of whatever it was estimated against before, which sounds like a mixture of symptomatic and asymptomatic disease. It would be helpful if the mismeasurement of eligibility was explored within levels of confounders as well as outcome/exposure. As predicted by Lewnard et al. this has led to bias away from the null. <https://pubmed.ncbi.nlm.nih.gov/34001753/>. Other considerations about the action of this bias in case-control samples were discussed by Jurek et al. <https://doi.org/10.1016/j.annepidem.2012.12.007>.

Response: We agree that this issue could also be described as selection bias. Notably, our study is also subject to potential selection bias associated with response to the questionnaire. To avoid potential confusion, we feel that is best to limit use of the term ‘selection bias’ to the latter phenomenon, and retain the phrase ‘outcome misclassification’ for the section on symptomatic status. Note that we clarify the nature of the bias at the start of the section (*“When asking individuals in the questionnaire to report their symptomatic status and comparing to SGSS for the assessment of outcome misclassification through symptomatic status...”*), aiding readers in interpretation of the ensuing results. We have added a sentence in the methods to show how this could have impacted selection into the original study: *“This could also have affected selection of the study population, since only symptomatic individuals were eligible for inclusion.”*

As suggested by the review, we have added a new Supplementary Table 6, which explores the association between key confounders and outcome misclassification. We describe these comparisons as follows:

Methods:

- *“To explore the potential mismeasurement of outcome misclassification within levels of confounders we described key confounders (as above) amongst those self-reporting asymptomatic versus*

symptomatic status. These were compared using percentage difference (with +/-5% set as threshold) and Chi-squared/Fisher's exact test."

Results:

- *"Self-reported asymptomatic status was associated with older age and male sex, but no other key confounders (based on +/-5% percentage difference and p values of <0.05; Table S6)."*

7. Because this is selection bias it would be more logical to list this first – before exposure misclassification. If doing a probabilistic bias analysis, the biases are reconstructed in the reverse order in which they occurred, so thoughtful consideration as to the order of bias is important.

Response: Thank you for this comment. As noted in the comment above, we prefer to apply the term selection bias to that associated with survey response, and have therefore opted to retain the wording around outcome misclassification.

8. Lines 133-4: the cross-reference for the relevant table or figure hasn't worked.

Response: We have corrected this error to ensure that Figure 2 is correctly cited.

9. Outcome misclassification, lines 136-146: Again this section is mislabelled. What is symptom onset date used for? It can be used as a restriction criterion for recruitment (so is a sampling/selection issue), or it can be used to measure the exposure if estimating waning VE (so is an exposure measurement error). Both impacts should be explored in the data. It is not acceptable to assume that there will be minimal difference. It has been shown that small differences can impact effect estimates. E.g. <https://doi.org/10.1093/ije/dym291> (other examples exist).

Response: As noted in response to comment 6, our preference is to retain the wording around outcome misclassification to reduce potential confusion with other selection biases affecting our study population, and we have instead added a clarification about the effect of outcome misclassification on study population selection (see point 6 above).

We agree that a more detailed quantification of the potential impact of symptom onset date uncertainty is warranted. We have re-estimated vaccine effectiveness based on self-reported symptom onset dates in the survey. These findings are presented as a new row in Figure 2, and are described in the text as follows:

Methods:

- *"Vaccination status using self-reported symptom onset date from the questionnaire was updated and amongst this population we reported vaccination status and ran the logistic regression models from the original study (see above)."*

Results: "

- *When updating onset dates to those self-reported in the questionnaire, vaccine effectiveness after two doses of BNT162b2 decreased from 88% (95% CI: 79-94%) to 87% (95% CI: 77-93%; Figure 2)."*

Discussion:

- *"Vaccination dates and symptom onset dates were consistent between the nationwide vaccination-COVID-19 PCR testing data (NIMS-SGSS) and the questionnaire with majority of individuals self-reporting the same date as in NIMS-SGSS. [...] For onset date, we likely underestimated misclassification since individuals were only asked to report their onset date in the questionnaire if different from the SGSS date that was provided. It is likely that some individuals could not recall the date and left this field blank, which would have been inaccurately determined as the correct date, rather than missing."*

Similarly, to the extra detail on vaccination date in response to comment 3, we have also added a histogram that shows the difference in days between self-reported and SGSS onset date (see Figure S3) as well as the number of individuals with a self-reported symptom onset date within 3 days +/- of the SGSS onset date. We have also provided the following text:

Methods:

- *“The [...] distribution in difference in days [was described] using a histogram. We also described the number of self-reported onset dates that were within 3 days +/- of SGSS (inclusive) or more and less than 3+/- days.”*

Results:

- *“95.8% of these were within 3 days +/- of SGSS date (inclusive), whereas 3.0% were more than 3 days earlier and 1.2% were more than three days later in the questionnaire (Figure S3).”*

Also, to align with the change in vaccination status in the above (comment 3), we have also provided the following text:

Methods:

- *“Vaccination status using self-reported symptom onset date from the questionnaire was updated and amongst this population we reported vaccination status [...]”.*

Results:

- *“When updating vaccination dates using self-reported onset dates, the percentage of non-vaccinated was very similar to when using SGSS (SGSS onset: 22.1%; self-reported onset: 22.6%).”*

In addition, we have provided a new Supplementary Table 7, which explores the association between key confounders and outcome misclassification. We describe these in the text as follows:

Methods:

- *“To explore the potential mismeasurement of outcome misclassification within levels of confounders we described key confounders (as above) amongst those self-reporting different versus same onset date in the questionnaire. These were compared using percentage difference (with +/-5% set as threshold) and Chi-squared/Fisher’s exact test.”*

Results:

- *“The prevalence of confounders did not differ among individuals with changed versus unchanged self-reported symptom onset date (based on +/-5% percentage difference and p values of <0.05; Table S7).”*

10. Confounding, lines 148-157: one of the key confounders in COVID-19 VE studies has been eligibility for vaccination and this highly conflated with occupation. E.g. in most countries, healthcare workers were prioritised for vaccination because they were considered to be at higher risk of infection. However, I don’t see this in the DAG or the list of confounders explored in the survey. Why not?

Response: We agree with the reviewer regarding the relevance of factors such as occupation. Notably, because this study focused on 70+ year olds, we assumed that most of the population did not have an occupation during the study period. We now clarify this in the Discussion as follows: *“Factors such as occupation may also be key confounders in younger adults, but were assumed to be less relevant in the current study given our focus on individuals over 70 years of age.”*

11. Confounders are all grouped together in the DAG, but this is unrealistic because they can be dependent on other nodes and the confounding caused may be direct or via other nodes to create backdoor paths. The DAG may need to be exploded to show each (or just a selection) confounder’s many paths.

Response: We thank the reviewer for this comment. We have updated Supplementary Figure 1 to provide a comprehensive overview of the key pathways under investigation in the current study. As noted in our response above (comment 4), we do not view this as a complete DAG, and clarify this in the figure title and legend (*“It should be noted that not all possible pathways are represented in the below figure”*). Our view is that this updated figure offers a clear and succinct summary of the overarching framework of our analysis, and is therefore preferable to confounder-specific DAGs.

12. Healthy vaccinee bias, lines 159-172: For clarity, the authors should state up front what kind of bias they think this is. I don’t see it depicted in the DAG. It reads like this is a source of confounding bias and effect modification; it could also be causing some collider bias. Again, the authors have claimed that this is unlikely to affect their estimates, but it would not be too hard to actually check this in the data.

Response: This point is well taken. The bias we are describing here is the potential increase in COVID-19 cases among non-vaccinated individuals due to the deferral of vaccination. To reduce confusion with confounding from health-seeking behaviour/healthcare access, we have renamed this section ‘Deferral bias’ (nomenclature also used by Hitchings et al, Epidemiology 2022 and Vasileiou et al, Lancet 2021, among others).

For clarity we have made the below edits to the text:

Methods:

- *“Deferral bias³⁶⁻³⁸ is potentially introduced if individuals delay their vaccinations because they have a COVID-19 infection, COVID-19 like symptoms or have been recently exposed to COVID-19 (individuals in the UK are asked to delay their vaccine by 28 days if they contract COVID-19/2019; Figure S1). “*

Results:

- *“Individuals with COVID-19-like symptoms, recent exposure to COVID-19, or a positive SARS-CoV-2 test just before their vaccination date were recommended to defer their vaccination by 28 days according to government guidelines(NHS UK). This deferral has the potential to increase vaccine effectiveness estimates as individuals that defer their vaccination for this reason might go on to test positive for SARS-CoV-2 (inflating cases among unvaccinated individuals). In the current study, among...”*

As suggested, we have re-estimated vaccine effectiveness accounting for the potential impact of this deferral bias. These findings are presented as a new row in Figure 2, and are described in the text as follows:

Methods:

- *“To assess by how much deferral bias might be expected to increase vaccine effectiveness estimates, we ran the logistic regression models from the original study (see above) removing individuals that reported they delayed either 2-3 weeks or 4 weeks because of COVID-19/COVID-19 like symptoms. We also described the vaccination status at symptom onset date of those that deferred their vaccination 2-3 or 4 week because of COVID-19/COVID-19 like symptoms.”*

Results:

- *“When assessing for the potential impact of deferral bias, amongst those who didn’t delay vaccination because of COVID-19/COVID-19 like symptoms (N=8,396), vaccine effectiveness after two doses of BNT162b2 decreased from 88% (95% CI: 79-94%) to 81% (95% CI: 67-90%; Figure 2).”*

Discussion:

- *“Vaccine deferral because of COVID-19/COVID-19 symptoms was relatively common in the study. When we excluded individuals who deferred their vaccination because of COVID-19/COVID-19 like symptoms, vaccine effectiveness estimates decreased from the original estimate of 88% to 81% (with overlapping confidence intervals). A decrease in estimated effectiveness is expected given that this approach entails removing non-vaccinated individuals who received a positive COVID-19 test from the analysis. The effect of deferral bias appears to be modest and does not undermine conclusions from the original TNCC study regarding the high effectiveness of vaccines during the initial phases of implementation.”*

13. Riskier behaviour, lines 174-184: again, is this confounding bias? It reads like it is causing confounding and effect modification. In Figure 1 it is shown as an intermediate. So, why not look at its effect by doing a very simple mediation analysis to determine the direct and indirect effects?

Response: Thank you for this comment. We consider this to be an alternative causal pathway between vaccination and infection, since vaccination may promote increased mixing with individuals from other households and thus increased exposure. We have modified the relevant subheadings of the Methods and Results to clarify this (*“Potential alternative causal pathways in original TNCC study”*). We have also expanded the description of this issue in the Methods section as follows: *“If vaccinated individuals start mixing more with individuals outside of their household after being vaccinated, then the risk of contracting COVID-19 might increase in these individuals creating an “alternative causal pathway” from vaccination to infection (Figure S1). If increased mixing occurs at a faster rate compared to non-vaccinated individuals’ then this could lower vaccine effectiveness estimates compared to true estimates.”*

Our primary aim was to quantify the extent of behaviour changes (as summarised in Figure 3E) as well as the potential association between self-reported increased mixing and COVID-19 (reported in the Results section).

14. Vaccination transportation, lines 186-198: this is depicted as a mediator in the DAG in Figure S1. However, the description makes it sound more like a selection bias issue. Is this really a form of collider bias? The Methods don't describe the mechanism clearly. Because DAGs need to show temporality, transport to the vaccination centre precedes the exposure so should be an antecedent of vaccination, whereas travelling from is descendent, so there probably needs to be more than one node for transport in the DAG.

Response: Thank you for this comment. As above (comment 13), we consider these to represent alternative causal pathways, and therefore group them under a new subheading in the Methods and Results (*"Potential alternative causal pathways in original TNCC study"*).

We agree with the reviewer's interpretation of the sequencing of these events. In the updated Supplementary Figure 1, we more clearly specify this, with multiple nodes specifying transport to (upstream) and transport from the vaccination centres (downstream). In practice, these peri-vaccination exposures would all precede the induction of robust immunity, as evidenced by the comparable event rates for ~10 days after dose 1 in phase 3 randomised controlled trials (e.g. Polack et al, NEJM 2021). We clarify this in the Methods as follows: *"These pathways include a composite of events immediately before and after vaccination, though in practice all exposures would precede the induction of robust vaccine immunity."*

Our primary aim was to quantify differences in transport route (as summarised in Figure 3F) as well as the potential association between riskier transport types and COVID-19 (reported in the Results section), and we have therefore retained our original approach.

15. It would be really nice if the authors took all these forms of bias and did a probabilistic (or deterministic) sensitivity analysis. Modern Epidemiology 4 has a nice primer on this (chapter on bias analysis) and there is also one in in "Applying Quantitative Bias Analysis to Epidemiologic Data" by Tim Lash and co. There is a package in Stata by Nicola Orsini that makes implementation a little more straightforward (<https://journals.sagepub.com/doi/pdf/10.1177/1536867X0800800103>). There is probably also one for R. Of course, this needs to also acknowledge limitations of bias analysis, e.g. <https://doi.org/10.1093/aje/kwab069>.

Response: Thank you for this suggestion. We agree that an analysis that incorporates all measured sources of bias is a valuable addition to the manuscript, and have added this. Our view is that the clearest way to do this is by adopting the existing analytic framework of updating the TNCC VE estimates accounting for all potential sources of bias simultaneously. We report on these updates analyses as follows:

Abstract:

- *"Using information from the questionnaire to produce a combined estimate that accounted for all potential biases decreased the original vaccine effectiveness estimate after two doses of BNT162b2 from 88% (95% CI: 79-94%) to 85% (95% CI: 68-94%)."*

Methods:

- *"Combined estimate accounting for all potential biases
When accounting for all biases at once, we ran the logistic regression models from the original study (see above) amongst those that did not delay their vaccination because of COVID-19/COVID-19 like symptoms, that self-reported they were symptomatic and using vaccination and symptom onset dates from the questionnaire adjusting for CEV, household size and type (as well as confounders adjusted for in the original TND study; Figure 2)."*

Results:

- *"Combined estimate account for all potential biases
When accounting for all of the above potential biases in the original TNCC study, vaccine effectiveness after two doses of BNT162b2 decreased slightly from 88% (95% CI: 79-94%) to 85% (95% CI: 68-94%; Figure 2)."*

Discussion:

- *"Using information from the questionnaire to produce a combined estimate that accounted for all potential biases decreased the original vaccine effectiveness estimate after two doses of BNT162b2 from 88% to 85% (with overlapping confidence intervals)." ... "The combined vaccine effectiveness*

estimate that accounted for all potential biases saw a modest decrease in effectiveness from the original estimate of 88% to 85% (with overlapping confidence intervals). Although this small change is reassuring, 85% should not be considered a best estimate since questionnaire responses that were provided in some cases many months after the events occurred cannot be considered the gold standard.”

16. The conclusions about the degree to which certain forms of bias were important or not would be easier to accept if the additional suggested analyses were performed to confirm the authors’ suspicions.

Response: As detailed above, we have extended the results to systematically re-calculate vaccine effectiveness accounting for each source of bias individually and all sources combined (comment 15). We believe this to be a more robust approach that lends stronger support to our stated conclusion, and we thank the reviewer for the valuable feedback.

17. The Discussion does not address the sampling bias that may be present in this study. There have been several papers discussing the utility of validation samples. I don’t see any of these cited in the discussion. E.g. <https://doi.org/10.1093/ije/dyy138> , [https://doi.org/10.1016/0895-4356\(88\)90020-0](https://doi.org/10.1016/0895-4356(88)90020-0),

Response: We acknowledge that there may be multiple sources of sampling bias at work in our study population, and thank the reviewer for pointing out that this was not made clear enough. We directly compare key characteristics of survey respondents and non-respondents in Supplementary Table 2, and describe key differences in the text as follows:

*“When comparing respondents with non-respondents of the questionnaire there did appear to be some demographic and clinical differences, with respondent being younger, more likely to be of White ethnicity, less likely to live in a deprived area, and more likely to be a case when compared with non-respondents (based on a percentage difference of +/-5% and p values of <0.05; **Table S2**).”*

In spite of these differences, we observed very similar VE estimates in respondents and non-respondents (as shown in Figure 2). As such, for the purposes of estimating VE, we do not anticipate selection bias from survey response to be a significant issue in our study. Nonetheless, key bias within the original TNCC population may also have affected VE estimates. We now address this issue in the Discussion as follows, citing the suggested reference by Infante-Rivard and Cusson (Int J Epidemiol 2018):

“Another limitation was that we were unable to assess whether collider bias³⁰⁻³² was present in the original study. Collider bias, another form of selection bias, could have potentially been introduced through the test negative design. This type of bias is potentially introduced as health-seeking behaviour is associated with testing, vaccination uptake and infection i.e., testing is a ‘collider’ on the pathway between vaccination and infection^{30,31,33}. We could not determine the presence of this bias because the association between health-seeking behaviour and testing could not be assessed as this information is not recorded in the data. Future studies should collect information on health-seeking behaviour so that this association can be assessed.

18. In the Methods, can it be clarified whether respondents were reminded of the period that was relevant to them and asked to only think about the behaviours in that period?

Response: The questionnaire referred to behaviours in the 3–4 weeks after their first and second vaccination. We have clarified this in the Methods as follows: *“To assess for riskier behaviour after vaccination the proportion of those that reported that they mixed the same, more or less in the 3–4 weeks after the date of their first or second vaccination was reported.”* We provide the full survey in the Supplementary appendix.

Regarding symptom status and onset date, the specific COVID-19 test date of interest was provided in the survey. We now clarify this as follows: *“The COVID-19 testing and symptom date of interest were specified in the survey letter (**Materials S1**).”*

19. I didn’t see anywhere the statistical program that was used for analysis. Was it R? Stata? SAS? Something else? Apologies if I missed this detail.

Response: As suggested, we now clarify these details the Methods as follows: *“All of the analyses were conducted using STATA (version 17) and R (version 4.1.3).”*

REVIEWERS' COMMENTS

Reviewer #2 (Remarks to the Author):

This manuscript is much improved. I was surprised to see that the biases explored did not appear to harm estimates. However, the addition of sensitivity analyses to confirm this lends greater credibility to the conclusions.

Some very minor comments that can be addressed at the Editor's discretion:

Be careful when using percentages that you are clear that you mean an absolute percentage point difference or a relative percentage change. E.g. a 5% difference should be clearly described as a 5 percentage point difference if that was what was meant (difference between 10% and 15%).

Be careful about statements like "(with overlapping confidence intervals)". Overlap (or not) of confidence intervals from two different models and two different datasets is not evidence of statistical significance. See: <https://doi.org/10.1007/s10654-016-0149-3>

Stata is a name, not an acronym. It does not need to be in all caps. (they are pretty emphatic about this on the stalist <https://www.stata.com/stalist/archive/2004-02/msg00429.html>).

REVIEWER COMMENTS

Reviewer #2 (Remarks to the Author):

This manuscript is much improved. I was surprised to see that the biases explored did not appear to harm estimates. However, the addition of sensitivity analyses to confirm this lends greater credibility to the conclusions.

Response: thank you very much for this positive feedback. We also agree that the paper is now much improved.

Some very minor comments that can be addressed at the Editor's discretion:

Be careful when using percentages that you are clear that you mean an absolute percentage point difference or a relative percentage change. E.g. a 5% difference should be clearly described as a 5 percentage point difference if that was what was meant (difference between 10% and 15%).

Response: We agree with this comment and therefore have provided more detail referring to absolute estimates throughout the methods section and in the headers of all tables in the main text and supplementary information.

Be careful about statements like "(with overlapping confidence intervals)". Overlap (or not) of confidence intervals from two different models and two different datasets is not evidence of statistical significance.

See: <https://doi.org/10.1007/s10654-016-0149-3>

Response: We agree with this comment and therefore have removed all references to overlapping point estimates throughout the whole manuscript.

Stata is a name, not an acronym. It does not need to be in all caps. (they are pretty emphatic about this on the statalist <https://www.stata.com/statalist/archive/2004-02/msg00429.html>).

Response: Thank you for pointing this out. We have edited the wording to be lower case.